# A tree of leaves: Phylogeny and historical biogeography of the leaf insects (Phasmatodea: Phylliidae)

Sarah Bank [1✉], Royce T. Cumming [2,3,4✉], Yunchang Li[1,5], Katharina Henze[1], Stéphane Le Tirant[2] & Sven Bradler [1]

The insect order Phasmatodea is known for large slender insects masquerading as twigs or bark. In contrast to these so-called stick insects, the subordinated clade of leaf insects (Phylliidae) are dorso-ventrally flattened and therefore resemble leaves in a unique way. Here we show that the origin of extant leaf insects lies in the Australasian/Pacific region with subsequent dispersal westwards to mainland Asia and colonisation of most Southeast Asian landmasses. We further hypothesise that the clade originated in the Early Eocene after the emergence of angiosperm-dominated rainforests. The genus *Phyllium* to which most of the ~100 described species pertain is recovered as paraphyletic and its three non-nominate subgenera are recovered as distinct, monophyletic groups and are consequently elevated to genus rank. This first phylogeny covering all major phylliid groups provides the basis for future studies on their taxonomy and a framework to unveil more of their cryptic and underestimated diversity.

[1] Department for Animal Evolution and Biodiversity, Johann-Friedrich-Blumenbach Institute of Zoology and Anthropology, University of Göttingen, Göttingen, Germany. [2] Montréal Insectarium, Montréal, QC, Canada. [3] Richard Gilder Graduate School, American Museum of Natural History, New York, NY, USA. [4] The Graduate Center, City University, New York, NY, USA. [5] Present address: Integrative Cancer Center & Cancer Clinical Research Center, Sichuan Cancer Hospital & Institute Sichuan Cancer Center, School of Medicine, University of Electronic Science and Technology of China, Chengdu, P.R. China. ✉email: sbank.bio@gmail.com; phylliidae.walkingleaf@gmail.com

The numerous defensive strategies that evolved in response to predation pressure are astounding and particularly diverse in insects. Many commonly known predator–prey interactions involve defence tactics that actively deter an attack, for instance by active escape, counter-attack or deimatism[1–3]. However, the primary defensive mechanism is to avoid detection itself. Being misidentified as an inedible item by a visually hunting predator and therefore reducing predation risk altogether may be achieved by masquerading as plant parts[4–6]. Although comparatively rare[7,8], such adaptations have evolved repeatedly among insects, for instance in butterflies, grasshoppers or mantises[9]. Among the most prominent examples are stick and leaf insects, an entire lineage of plant mimics referred to as the insect order Phasmatodea.

The majority of the over 3,000 described species of Phasmatodea exhibit slender, elongated body forms resembling twigs[10]. Several lineages have independently developed additional morphological structures to conceal themselves in other habitats aside from branches and foliage such as moss, lichen, leaf litter or bark. A leaf-like habitus is however rather rare and generally considered the most elaborate plant masquerade[11,12]. While plant and gymnosperm leaf mimicry has been documented for insects from as early as the Middle Permian[13] and more frequently from the Mesozoic[9,12,14], the simulation of angiosperm leaves is a phenomenon that arose as a result of the recent diversification of flowering plants. Hence, these adaptations appear to only occur in few insect orders[15–18], one of them being the phasmatodean lineage of leaf insects or walking leaves (Phylliidae).

Distributed across the tropical regions of Asia, Australasia and the Pacific, Phylliidae uniquely exhibit a nearly impeccable leaf masquerade accomplished by a dorso-ventrally flattened body form with a leaf-like venation pattern and lobe-like extensions on the abdomen and legs (Fig. 1). Although predominantly green, leaf insects show a considerable diversity in colour and pattern representing different stages of leaf decay (Fig. 1b-d). However, colouration appears to be a response to specific environmental conditions (i.e., phenotypic plasticity) and may vary between conspecifics (see Fig. 1a, b). Males can be easily distinguished from females by several pronounced dimorphic traits (Fig. 1e, f). Besides being larger, females have reduced hind wings but enlarged tegmina covering most of the abdomen, whereas males possess fully-developed hind wings and shorter tegmina[19]. The capability of active flight in males along with the presence of long antennae appears to play a vital role in mate search, while the inconspicuous female may use its short antennae for defensive stridulation[1,20]. Other behavioural adaptations revolve around perfecting leaf masquerade in the inactive phase during the daytime. While phylliids are mostly found in a motionless posture (adaptive stillness) with their head resting in a notch formed by the fore femora (Fig. 1 a-c), disturbance may trigger a swaying motion simulating the movement of leaves in light wind[1,21].

The distinctness of leaf insects from the remaining phasmatodeans is indisputable and led to the designation of a separate order (Phyllioptera) as sister taxon to all other phasmatodeans in the past[22]. Although all phylogenetic studies agree on Phylliidae as a member of the Euphasmatodea (=Phasmatodea excl. *Timema*), its phylogenetic position has long remained unclear. Zompro[23] proposed Phylliidae as a sister group to all other Euphasmatodea (excl. *Agathemera* = Verophasmatodea therein), while more recent molecular analyses recovered them as sister to

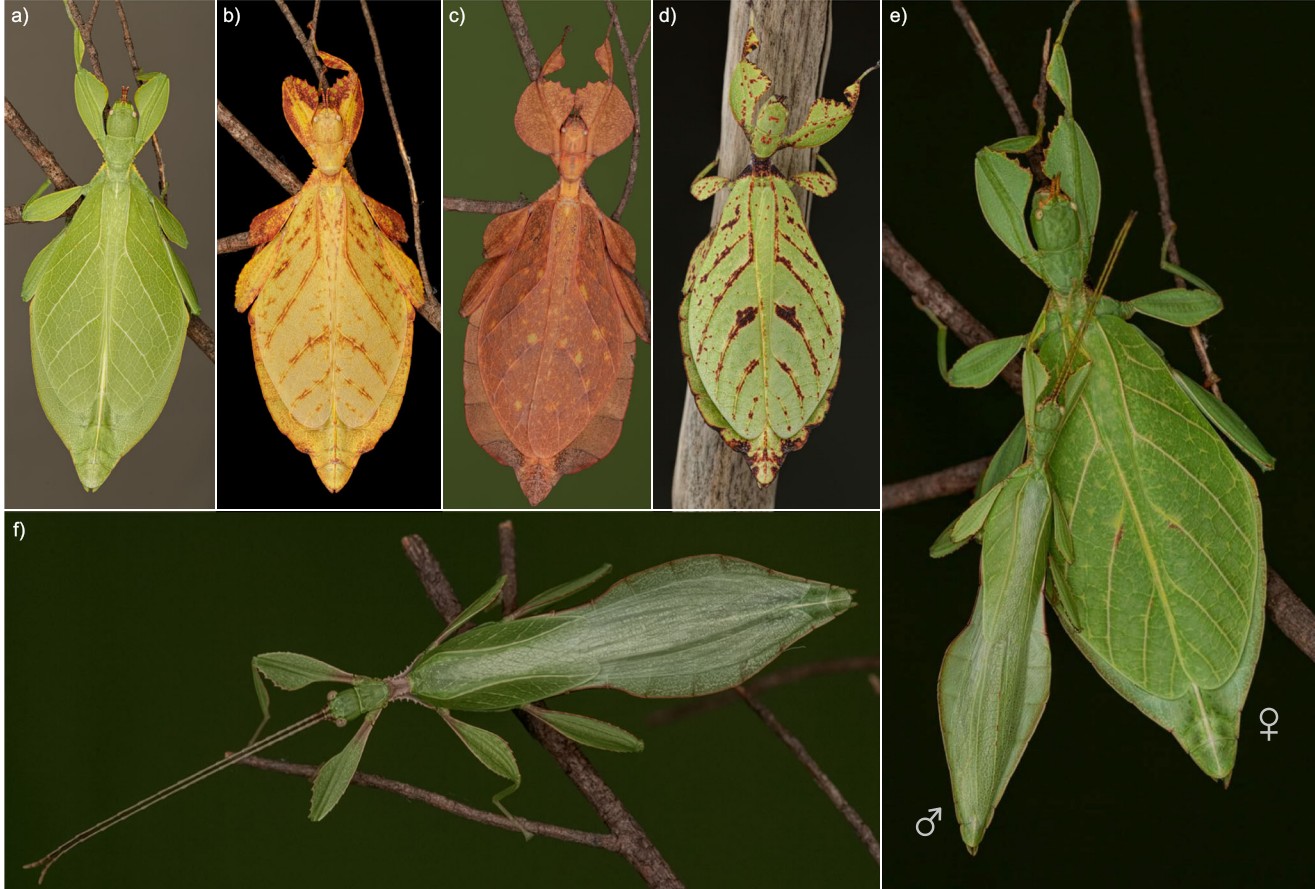

**Fig. 1 Photographs of leaf insect specimens (Phylliidae). a,b** phenotypic variations of *Phyllium elegans* females, **c** female of *Phyllium letiranti*, **d** female of *Phyllium hausleithneri*, **e** couple of *Phyllium rubrum* and (**f**) male of *Phyllium letiranti*. Photographs by Bruno Kneubühler.

the remaining Neophasmatodea (=Euphasmatodea excl. Aschiphasmatidae)[24–26]. However, the majority of phylogenetic studies based on morphological, genetic and even genomic data recover leaf insects as a rather subordinate lineage within Euphasmatodea[19,27–34]. The sister group of leaf insects remains a matter of debate, whereas their internal relationships have never been the subject of a comprehensive analysis.

While Bradler and Buckley[10] noted that Phylliidae only account for <2% of the phasmatodean diversity with about 50 known species, the number of described species has now doubled within just a few years. Misidentification, overestimation of species' distributions and the unreliability of the highly variable morphological traits[35] had resulted in a chaotic taxonomy that only recently started to be overcome by extensive morphological examinations (e.g., Cumming et al.[36,37]). Captive breeding and molecular analysis have further helped to shed light on the phylogenetic relationships and to match up males and females of leaf insects[38–40]. According to the most recent studies, Phylliidae currently includes six genera (*Chitoniscus*, *Cryptophyllium*, *Microphyllium*, *Nanophyllium*, *Phyllium* and *Pseudomicrophyllium*) with most species pertaining to *Phyllium*, which is further divided into four subgenera (*Comptaphyllium*, *Phyllium*, *Pulchriphyllium* and *Walaphyllium*). Both *Phyllium* and *Pulchriphyllium* have undergone further intra-generic systematisation and were classed in several species groups by Hennemann et al.[35]. One of these species groups was recently revealed to be distinct to the remaining phylliids and was therefore transferred to the newly erected genus *Cryptophyllium*[40]. Molecular analyses preceding this study had already repeatedly demonstrated that *Phyllium* (and *Chitoniscus*) are paraphyletic and that the Phylliidae are in need of revision[24,25,29,40].

Here, we present the first phylogeny covering all major phylliid lineages and confirm the paraphyly of the genera *Chitoniscus* and *Phyllium*. Based on our results, we were able to render *Phyllium* monophyletic by elevating its monophyletic subgenera to the rank of genus. Our divergence time estimation and reconstruction of the group's historical biogeography suggest an origin of extant Phylliidae in the Australasian/Pacific region in the Palaeogene. Subsequent dispersal and radiation are discussed in light of the co-evolution with angiosperms.

## Results and discussion

**Phylogeny and systematics.** For 77% of all analysed taxa, we obtained the sequences of five or six genes and for 3% of the included taxa we could generate sequences of only one or two genes (for further details, see Supplementary Data 1). Both ML and BI phylogenetic analyses have produced mostly congruent phylogenies with comparable support values (Fig. 2, Supplementary Figs. 1–5). The outgroup taxa adapted from Bank et al.[26] were found to present a similar topology with minor differences in regard to weakly supported sister group relationships (Supplementary Fig. 1). The Neophasmatodea are maximally supported and all clades with the exception of Bacillinae were recovered with reliably high node support (i.e., UFBoot >95%; posterior probability (PP) and SH-alrt >80%). Standard nonparametric bootstrap (BS) values were found to be generally lower, but also to support the majority of taxa. The weakly supported deeper nodes are consistent with the assumption of a neophasmatodean rapid radiation that is unresolvable using a limited set of loci[25].

Phylliidae are recovered in all trees as the sister group to the remaining Neophasmatodea, a topology that was also obtained in some previous studies based on just seven or fewer loci[24–26]. This might actually be artificial due to a bias resulting from the similarity of these loci to Aschiphasmatidae, which were recently

estimated to be the sister group to Phylliidae[41]. In contrast, mitogenomic studies showed Phylliidae as an early diverging lineage closely related to Lonchodinae[42–46], whereas phylotranscriptomic analyses recovered them as deeply nested within the Oriophasmata[33,34]. While both transcriptomic studies were based on the same dataset, only Simon et al.[33] recovered the Phylliidae as the sister group to the European *Bacillus* and related to Malagasy taxa. The ensuing inclusion of Bacillinae specimens in our analysis, however, could only confirm the close relationship of Bacillinae with the Malagasy stick insects[25,33,34,41]. Although the subordinate placement among Oriophasmata can be considered more conclusive due to the larger amount of data, it is noteworthy that only a single phylliid species (*Phyllium philippinicum*) was actually included. Thus, the sister group of Phylliidae still remains uncertain and requires further investigation in a phylogenomic context including several leaf insect species and more outgroup representatives.

All our phylogenetic inferences corroborate the monophyly of Phylliidae with maximum support (Fig. 2 and Supplementary Figs. 2–5). However, *Chitoniscus* and *Phyllium* are recovered as paraphyletic, which was already shown in previous studies based on molecular data[24,25,29,40,41]. The *Chitoniscus* spp. from the Fiji islands and from New Caledonia are found to be distinct, unrelated clades on whose taxonomical status we cannot elaborate without the inclusion of the type species *C. lobiventris* (Fiji). In contrast, the *Phyllium* type species (*Phyllium (Phyllium) siccifolium*) could be confirmed as a member of *Phyllium (Phyllium)*. Moreover, the distinct yet monophyletic and highly supported clades of polyphyletic *Phyllium* are in fact corresponding to its four subgenera (with the exception of two species, see below). Hence, in order to render *Phyllium* monophyletic, we elevated the remaining subgenera to genus rank (see the new classification in Fig. 2; please refer to Supplementary Discussion and Supplementary Notes 1, 2 for more details on taxonomical acts and an identification key, and to Supplementary Data 2 for a checklist of all phylliid species). Two species originally assigned to *Phyllium* were not found to belong to any of the former *Phyllium* groups: *Phyllium (Pulchriphyllium) brevipenne* and *Phyllium (Phyllium) geryon*. The former had previously been suggested to be closely related to the *frondosum* species group[35], a clade that was recently revealed to belong to *Nanophyllium*[39]. As our phylogenetic inferences recover *P. brevipenne* as the sister taxon to *Nanophyllium*, the species is hereby transferred to *Nanophyllium* as *Nanophyllium brevipenne* **comb. nov.** (Supplementary Discussion). The second species, *P. geryon*, was recovered as the sister group to *Pseudomicrophyllium*. While both *Microphyllium* and *Pseudomicrophyllium* are mainly distinguished from the remaining genera by their smaller size, we were able to identify several morphological characteristics to link the larger *P. geryon* to *Pseudomicrophyllium*, which is therefore transferred to *Pseudomicrophyllium* as *Pseudomicrophyllium geryon* **comb. nov.** Moreover, we recovered the Sri Lankan population of *Pulchriphyllium bioculatum* (subspecies *agathyrsus*) as unrelated to *Pu. bioculatum*, which prompted us to reinstate its former status as a full species (Supplementary Discussion).

Of the five *Phyllium* species available on GenBank, which were not processed by and included in Cumming et al.[38,40], only one identification could be confirmed as correct (*P. giganteum*). The *Phyllium celebicum* from Thailand[29] was identified and bred in captivity as such, despite the fact that the type species is from Sulawesi. Here, we recovered the specimen as the sister group to *Cryptophyllium westwoodii*, which is also distributed in Thailand. Similarly, the female *Phyllium* sp. from Papua New Guinea[47] was found to be nested within the Australasian *Nanophyllium*. The specimen was probably believed to be a member of the *frondosum* species group (see above), whose species were recently transferred

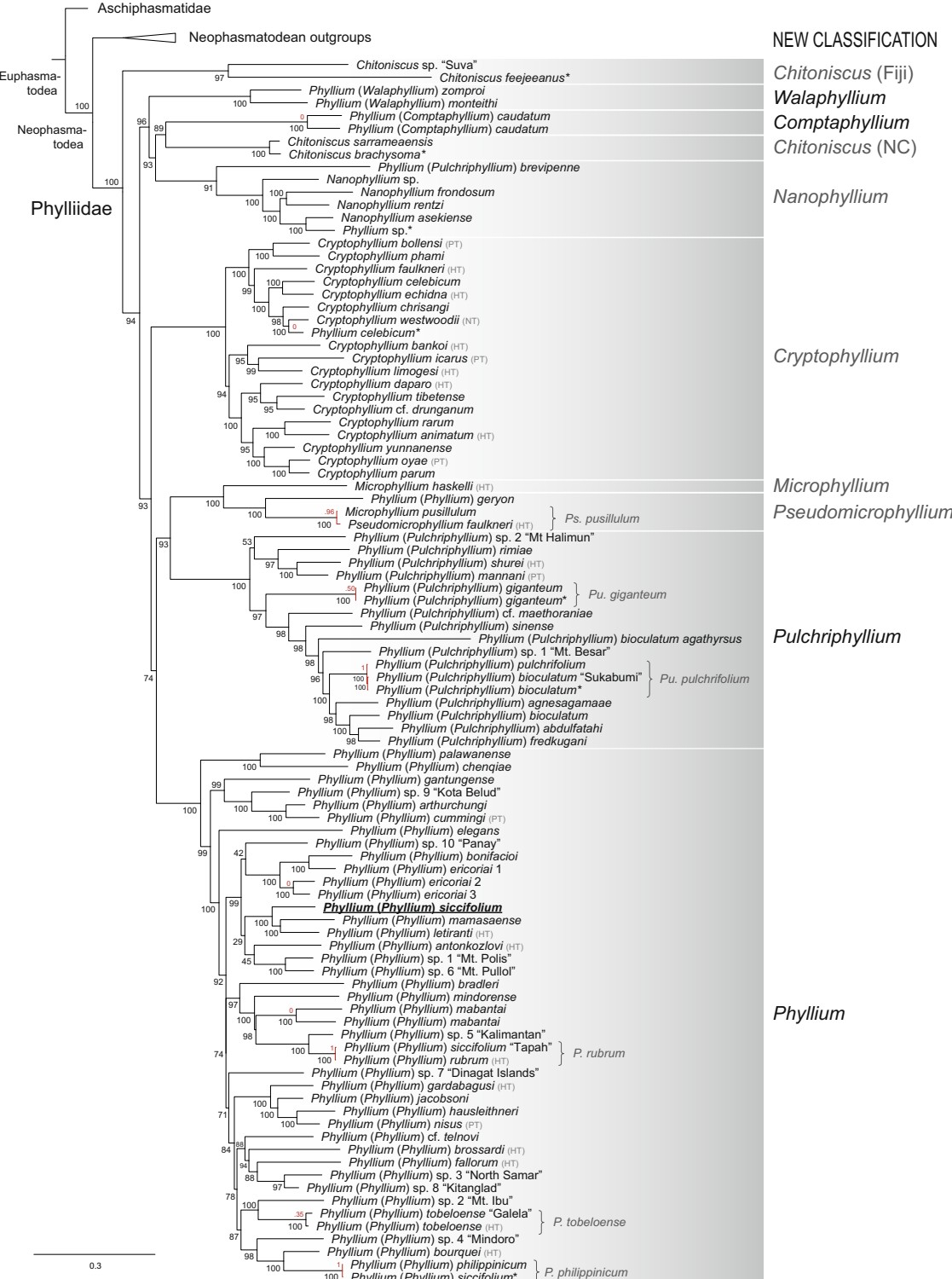

**Fig. 2 Maximum likelihood (ML) phylogenetic relationships of Phylliidae.** The topology is derived from the best-scoring ML tree in IQ-TREE using six nuclear and mitochondrial loci. UFBoot node support values are depicted at each node. The probability values resulting from the species delimitation analysis are shown in red (red branches if specimens were estimated to be the same species). The new generic classification following our results is presented on the right. The *Phyllium* type species (*P. siccifolium*) is underlined and noted in bold. HT holotype, NT neotype, PT paratype, NC New Caledonia. *Data solely obtained from GenBank.

to *Nanophyllium*. The limited number of unambiguous morphological characteristics impedes accurate identification, particularly if type material is unavailable. The cryptic diversity of leaf insects constitutes yet another problem: Several specimens that are morphologically indistinguishable are recovered as distinct species in the molecular phylogeny (e.g. *P. ericoriai*, *P. mabantai*; see also Cumming et al.[40]). However, only the inclusion of type material in the molecular analysis allows to reveal which specimens can be assigned to a described species. Hence, in addition to the 15 undescribed phylliid species, we present five other putative new

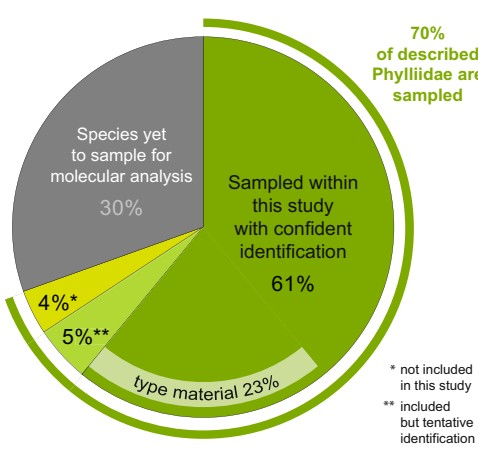

**Fig. 3 Pie chart of species coverage.** The majority of Phylliidae are sampled with 66% of described species covered in this study, including type material of 24 species and 11 non-type specimens from the type localities (Supplementary Data 2). While the material for 30% of described species was not available, sequence data for 15–20 additional and putatively new species were included in our study.

species, which the species delimitation analysis determined as distinct (Fig. 2). Future studies need to explore the potential of the putative new species and clarify their status.

**Phylogeny and evolution of Phylliidae.** Our taxon sampling covers the majority of described species (~66%), one-third of which are type material (Fig. 3). The species coverage per genus lies above 50% for seven of the nine genera, with the species-rich *Cryptophyllium*, *Phyllium* and *Pulchriphyllium* being represented by 75%, 84% and 63% of the described species, respectively. The low species coverage of the Australasian taxa suggests a sampling bias between these regions and SE Asia. The high amount of putative new species as well as cryptic species points to the conclusion that species diversity must be assumed to be highly underestimated, in particular for the Australasian and Pacific islands.

Our comprehensive taxon sampling of leaf insects and the combined usage of molecular and morphological data allow new insights on their phylogenetic relationships involving all genera. As stated above, all genera are recovered as distinct clades in both ML and BI inferences. In contrast to morphological studies[11,23,48], where *Nanophyllium* as the only member of the tribe Nanophylliini[48] was hypothesised as the sister group to the remaining phylliids (Phylliini), our results reveal the genus to be a subordinate clade within Phylliidae. The proposition of *Nanophyllium* as a high-ranking phylliid clade was based solely on males, which bear morphologically unique characteristics[39,48]. The recent unveiling of *Nanophyllium* females (as already described species within *Pulchriphyllium*[39]) indicated a potential closer relationship of *Nanophyllium* to one of the Phylliini (sub) genera and our inferences corroborate that the tribal subdivision does not reflect the phylogenetic relationships of Phylliidae. Instead of *Nanophyllium*, we recover the *Chitoniscus* from Fiji as the earliest diverging taxon as was shown by previous molecular analysis[24,25,29,41]. Corresponding to the same studies, we also found that the *Chitoniscus* from New Caledonia (NC) are unrelated to the Fiji lineage. In fact, our results concur according to a geographical pattern and reveal a lineage with the Australasian/Pacific distribution that forms the sister group to the remaining phylliids. Within this clade, *Chitoniscus* (NC) are found as the closest relative to *Comptaphyllium*, while it remains uncertain whether *Walaphyllium* is the sister group to

*Nanophyllium* + (*Chitoniscus* + *Comptaphyllium*) (ML tree) or is more closely related to *Nanophyllium* (BI tree). Regarding the remaining phylliid taxa, BI and ML results show an incongruence concerning *Cryptophyllium*, which is recovered as the sister group to the remaining lineages under ML and as more closely related to *Pulchriphyllium* + (*Microphyllium* + *Pseudomicrophyllium*) under BI. However, both hypotheses are only weakly supported and thus the position of *Cryptophyllium* remains unclear. Despite the uncertain relationship of the aforementioned two lineages, our phylogeny based on an increased taxon sampling is largely robust and provides a sound basis for future studies.

**Divergence times and the evolution of the leaf habitus.** Both BEAST analyses converged and resulted in identical topologies except for the weakly supported positions of *Phyllium mindorense* and the clade of *P. siccifolium* + (*P. mamasaense* + *P. letiranti*) (Supplementary Figs. 4 and 5). Divergence times are largely congruent, with the origin of Phasmatodea estimated at ~73.8 million years ago (mya) (90.2–58.9 mya) for the fossil-calibrated (FC) tree and ~77.8 mya (89.7–65.73 mya) for the root calibration (RC) with estimates derived from Simon et al.[33]. The divergence of Phylliidae was estimated to have started at ~49.9 mya (55.5 – 47.1 mya) and at ~51.1 mya (64.0–38.2 mya) for FC and RC analyses, respectively, with the clades established as genera largely originating in the Oligocene. While our estimates are comparable to previously obtained divergence times[25,33] and within the credibility intervals of others[24,41], the analyses by Tihelka et al.[34] and Forni et al.[45] have presented a much older origin of Euphasmatodea (Jurassic) and Phylliidae (Cretaceous) (Fig. 4). The choice of unequivocal fossils and appropriate calibration points is essential and their inconsistent application may lead to substantial discrepancies among studies on phasmatodean evolution (but see previous discussions[10,26,49]).

The life history of stick and leaf insects was largely shaped by the co-evolution with land plants. Adapted to a tree-dwelling life style, phylliid masquerade is achieved by simulating the broad leaves of flowering plants and the additional imitation of the diffuse growth of leaf veins in the female forewing venation[19] that has perfected their cryptic appearance in the foliage. This uniform adaptation is best described as a nonadaptive radiation in which the diversification was not accompanied by relevant niche differentiation[50], resulting in taxa with little or no ecological and phenotypic variation[51], as has been recently suggested for a clade of uniformly ground-dwelling stick insects[26]. Both the broad leaf habitus and the net venation of Phylliidae are evidently linked to the eudicot angiosperm evolution. While it had been argued before that leaf mimicry predated the more common twig mimicry of extant forms, since fossil stem-Phasmatodea as well as members of Timematodea, the sister taxon of Euphasmatodea, exhibit leaf mimicry[14,52], it appears undisputed that phylliid leaf insects derived from twig-imitating forms[11] and secondarily evolved angiosperm leaf imitation more recently. While the origin of angiosperms is still under debate (see Silvestro et al.[53] and references therein), recent studies based on fossil and molecular data appear to agree on high diversification rates and radiation events of several families during the Cretaceous[53–58]. With the beginning of the Cenozoic, and in strong correlation with the gradual extinction of gymnosperms[59], angiosperms became increasingly dominant in most terrestrial ecosystems[53,56,58,60]. Our divergence time estimation places the origin of Phylliidae in the Early Eocene (i.e., Early Cenozoic) following the preceding burst of angiosperm diversification (Fig. 4). Although the abundance of flowering plants and their dominance within tropical rain forests should be regarded as a prerequisite for the evolution of leaf insects, recent studies[34,41,45] have challenged the Cenozoic origin of leaf insects and proposed an earlier

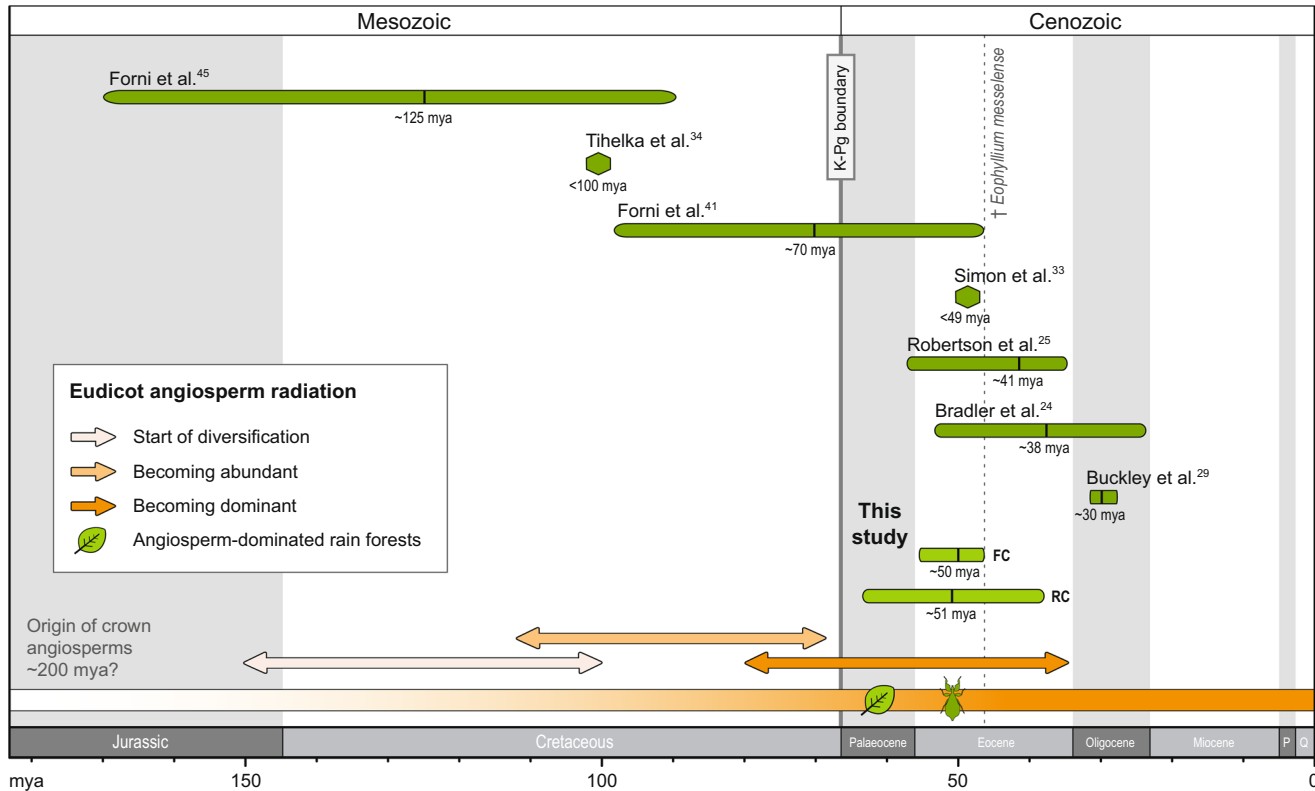

**Fig. 4 Geological timeline comparing the divergence time estimates for Phylliidae in relation to the angiosperm radiation.** Green confidence interval bars are given for each study that estimated a divergence time for leaf insects; hexagonal shapes are depicted when the estimated age is referring to the divergence from the phylliid sister taxon. The radiation of eudicot angiosperm is hypothesised to have started in the Early/Mid Cretaceous[54–57, 60] with extant lineages having become abundant until the Late Cretaceous. Dominance over gymnosperms and ferns was probably achieved during the period from the Late Cretaceous to the Early Cenozoic[53, 56] with subsequent emergence of angiosperm-dominated rain forests. FC fossil calibration (†*Eophyllium messelense*), RC root calibration (Euphasmatodea).

divergence in the Cretaceous or Jurassic. In particular, large parts of the lower ages estimated by Forni et al.[45] (approximately 170–90 mya) appear to be too old given that eudicot angiosperms are hypothesised to have been subordinate herbs until the mid-Cretaceous[55,61], a span of time only covered by the upper confidence interval in Forni et al.'s study[45]. The first forest trees may have occurred from that time on, but rainforests dominated by angiosperm trees probably arose at the end of the Cretaceous[61–64]. Interestingly, the origin of other leaf-mimicking insects such as members of the orthopteran Tettigoniidae[17,65] or the *Kallima* butterflies[16,66] appear to coincide with our age estimates for Phylliidae, supporting our claim that leaf masquerade involving angiosperm leaf imitation cannot have evolved at a time predating the angiosperm predominance.

**Historical biogeography**. Both BI trees (Supplementary Figs. 4 and 5) resulted in similar ages for the Phylliidae, with estimates differing by only 1 or 2 million years. We decided to carry out the ancestral range estimation based on the FC tree, mainly due to smaller confidence intervals. Despite the few weakly supported phylogenetic relationships, we deem it unlikely that our estimation is biased or negatively affected. The only crucial incongruence with the ML tree, the position of *Cryptophyllium*, appears to be irrelevant in regard to the biogeographical pattern, since the dispersal involving Borneo (Sundaland) must inevitably be assumed.

We found that the historical biogeography estimated in BioGeoBEARS was best represented with the highest likelihood under the DEC model (Fig. 5; see all results in Supplementary

Fig. 6). According to our analysis, extant Phylliidae originated in the Early Eocene (55.5–47.1 mya) in the Australasian/Pacific region (Fig. 6). Considering the Oriental origin of Oriophasmata[33] and that the fossil stem group leaf insect *Eophyllium messelense* was distributed in Europe[11], it is likely that the ancestors of crown group Phylliidae were distributed in Southern Eurasia. With the beginning of the Eocene, climatic changes and the continental collision of India led to an increased biotic migration towards the continuously tropical regions of SE Asia[67–70]. Being conserved in their climatic niche, leaf insects were most likely also influenced by these processes and dispersed in a similar pattern, as for instance reported for tropical spiders[70]. Based on our inference, the origin of extant Phylliidae is in the Southwest Pacific region, which suggests an oceanic long–distance dispersal from continental Asia (Sundaland). The geological history of this region is extremely complex and impedes adequate biogeographical reconstruction[71–73]. Therefore, we can only assume that extant leaf insects derived from a lineage that colonised a landmass in this region such as the proto-Papuan archipelago[74,75] or the South Caroline arc, which allowed subsequent dispersal to the Pacific Islands and to New Guinea.

Since the estimated divergence time of Fijian *Chitoniscus* (38.7–18.4 mya), which form the sister group to the remaining leaf insects, is coinciding with the emergence of Viti Levu[76], it is likely that the colonisation of Fiji occurred via the Vitiaz arc[72,77–79] (Fig. 6a). This chain of emergent volcanic islands facilitated dispersal from the Philippines over the South Caroline arc and the Solomon Islands to Fiji from the Early Oligocene on and was linked to the eastwards dispersal of other arthropod groups[26,80–83]. The only leaf insect from the Solomon Islands

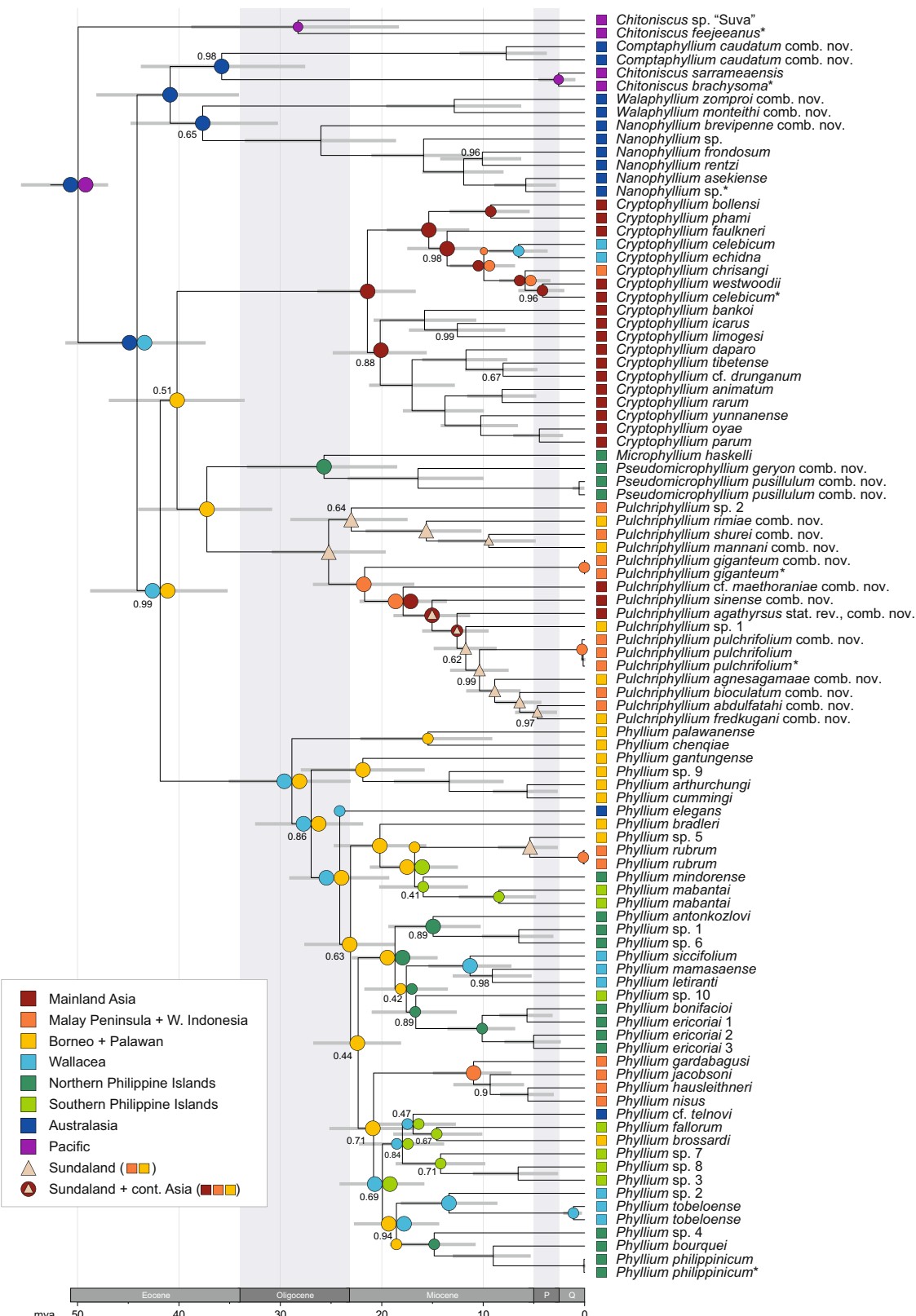

**Fig. 5 Dated Bayesian phylogeny and ancestral range estimation of Phylliidae.** The tree was derived from the fossil-calibrated BEAST analysis using †*Eophyllium messelense* (47 mya) (Supplementary Fig. 4). The 95% credibility intervals and node support values with <1 PP (posterior probability) are depicted at the nodes. The results of the biogeographical analysis under the DEC model are presented at each node with the colour code corresponding to the areas in the legend and to Fig. 6. Please refer to Supplementary Fig. 6 for a more detailed illustration of the relative probabilities of ancestral ranges. P Pliocene, Q Quarternary. *Data solely obtained from GenBank.

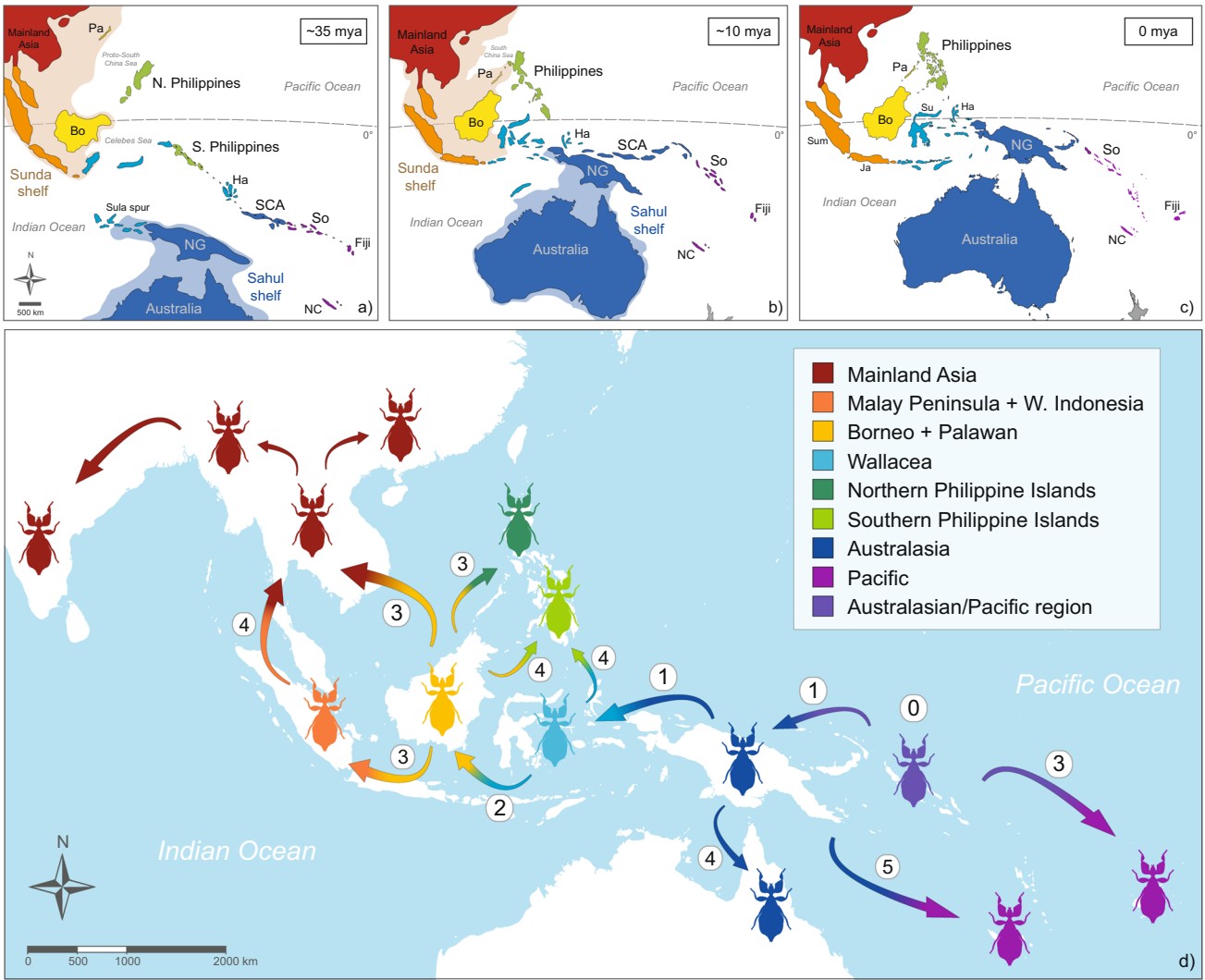

**Fig. 6 Schematic overview of the historical biogeography of Phylliidae. a–c** Palaeogeography of SE Asia and the SW Pacific (adapted from Hall[72]) showing the northward movement of the Sahul shelf and the drift of the fragments of the Philippines and Wallacea. The dotted line in (**a**) indicates which landmasses will be connected via the Vitiaz arc (~30 mya). Note that the depiction of landmasses does not necessarily imply that they were emergent at that time. Both Sunda and Sahul shelves may have been exposed as dry land during low sea–level stands. **d** Main dispersal events of extant leaf insects with origin in the Australasian/Pacific region (0). The numbering illustrates our suggested chronological order of colonisation. Colour code corresponds to Fig. 5. Bo Borneo, Ha Halmahera, Ja Java, Pa Palawan, NC New Caledonia, NG New Guinea, SCA South Caroline arc, So Solomon Islands, Su Sulawesi, Sum Sumatra.

(*Pulchriphyllium groesseri* **comb. nov.**, not included in this study) suggests a possible relationship with *Chitoniscus* (Fiji) as a remnant of the ancient migration to Fiji. Since the traditional taxonomy of Phylliidae is found to be unreliable, the possibility that this species might not belong to *Pulchriphyllium* is compelling. However, without material available for its inclusion in a phylogenetic analysis, we cannot exclude that *Pu. groesseri* colonised the Solomon Islands at a later time.

New Guinea or a related landmass being the source area for the dispersal to Fiji is further substantiated by the early diverging Australasian clade consisting of *Chitoniscus* (NC), *Comptaphyllium*, *Nanophyllium* and *Walaphyllium*. Their diversification started ~40.9 mya (48.1–34.2 mya), an estimate that is in fact not consistent with geological hypotheses concerning the more recent emergence of New Guinea[72,84,85]. However, our results suggest that a proto-New Guinean landmass was already emergent, probably a fragment of former island arc (Oliver et al.[79]), which is in line with several biogeographical studies that proposed similar hypotheses[74,86–90]. This also appears to correspond to New

Guinea's high endemism and biotic difference to (Eastern) Australia[91,92]. The lineages currently found on New Guinea, *Comptaphyllium* and *Nanophyllium*, diversified in the Oligocene and Miocene, a range of ages, which is in agreement with the diversification of other lineages such as butterflies and curculionid beetles[83,90,93–95]. Subsequent dispersal to Australia (*Walaphyllium*, 19.5–6.3 mya) and to New Caledonia (*Chitoniscus*, 4.5–1.0 mya) may have occurred more recently and may be explained by long–distance dispersal events.

A dispersal event to the West gave rise to the lineages of SE Asia including the Philippines and Wallacea. Our inference proposes an origin in the Late Eocene in Borneo/Wallacea, which we interpret as a transit zone to Sundaland. The clade comprising *Cryptophyllium*, *Microphyllium*, *Pseudomicrophyllium* and *Pulchriphyllium* split from *Phyllium* ~41.9 mya (48.68–35.3 mya), followed by its diversification in Borneo ~40.2 mya (46.8–33.6 mya). *Cryptophyllium* appears to have dispersed in the Early Miocene (26.3–16.8 mya) across Sundaland to mainland Asia before high sea-level stands disallowed the transgression of the Isthmus of Kra for a

prolonged period of time[96]. Their distribution expanded to India and Nepal as well as to Southern China[40]. Yet, one lineage migrated southward to Wallacea via Sundaland by dispersing over the narrowing Makassar Strait to Sulawesi (*C. celebicum* + *C. echidna*), corroborating the permeability of Wallace's Line in the Late Miocene[95,97]. Its sister group remained in Sundaland, splitting into *C. chrisangi* and *C. westwoodii*, whereas the latter dispersed northward before the transgression of the Isthmus of Kra was again limited in the Pliocene[96].

The sister group of *Cryptophyllium* is estimated to have a Bornean (Sundaland) origin in the Late Eocene and split into two lineages. While *Pulchriphyllium* appears to have diversified from the Late Oligocene on (30.8–19.71 mya) and clearly shows a Sundaland distribution with a few representatives nowadays found on mainland Asia and the islands of the Indian Ocean, a separate dispersal introduced the common ancestor of *Microphyllium* and *Pseudomicrophyllium* to the Northern Philippines at about the same time (33.2–18.6 mya). A migration via Palawan is unlikely, since Palawan was not associated with Borneo until the end of the Miocene (Fig. 6a, b). Alternatively, dispersal may have occurred via a volcanic island arc formed by different fragments of the Philippines (Luzon-Sulu-Sabah arc[72]); however, it is not certain that these islands were continuously emergent[84].

Our results suggest that *Phyllium* has its origin in Borneo/Wallacea in the Mid Oligocene (35.0–23.2 mya). However, due to the widening marine barrier of the Makassar Strait between Borneo and Sulawesi and the early divergent lineages that clearly diversified on Borneo, we favour a Bornean origin as suggested by the results based on the DIVALIKE or BAYAREALIKE biogeographic models (Supplementary Fig. 6c, d). A transition via Wallacea is however highly likely regarding the Australasian *P. elegans*, which split from its Bornean sister group in the Late Oligocene (29.1–19.4 mya) and reached New Guinea probably via long–distance dispersal across the Wallacean islands of the Banda arc and Sula Spur[84,98,99] (Fig. 6). From Borneo, several lineages have independently colonised the islands of the Philippines and Wallacea, as well as Western Indonesia and the Malay Peninsula. While the Philippines were most likely colonised from Borneo via the Sulu archipelago[100,101], the colonisation from Wallacea probably occurred across the island arc of the Sangihe-Talaud archipelago[26,99,100,102]. In contrast to the other leaf insect genera, *Phyllium* colonised multiple islands and regions, resulting in a high number of independent speciation events. Interestingly, most species appear to have originated in the Miocene, substantiating their early divergence from sister taxa and explaining the high incidence of endemic leaf insects across oceanic SE Asia.

## Methods

**Taxonomic sampling**. We selected 96 phylliid specimens with representatives of each genus and subgenus to be included in our phylogenetic analysis covering about two-thirds of the currently known diversity[103]. Molecular data for 37 species had already been published and were available on GenBank[29,38,40,47,104]. We chose to resample some of these due to the high probability of species having been misidentified in the past. For 29 of the recently published species by Cumming et al.[38,40], we used the voucher specimens to generate sequences for missing genes and data for 59 phylliid specimens were generated de novo. We used the same outgroup with representatives of each major phasmatodean lineage as outlined by Bank et al.[26] and added five of the Heteropterygidae species published therein. Because the African *Bacillus* had been inferred as the sister taxon to Phylliidae in the transcriptomic study by Simon et al.[33], we included five species of Bacillinae, adding up to 73 outgroup species and 169 specimens in total (see Supplementary Data 1 for more details).

**Molecular laboratory and phylogenetic analysis**. All specimens were either preserved in ethanol (70–100%) or dry-pinned prior to the removal of the femoral muscle tissue from the hind or mid leg. In a few cases, newly hatched nymphs were used. DNA extraction, PCR and sequencing followed the protocols outlined by Bank et al.[26]. While Bank et al.[26] targeted three nuclear and four mitochondrial markers (*18 S*, *28 S*, *H3*, and *COI*, *COII*, *12 S*, *16 S*), the amplification of the *12 S*

rRNA gene was repeatedly unsuccessful for phylliid samples, so we decided to exclude this locus. We deposited the newly obtained sequences in GenBank (Supplementary Data 1).

Multiple sequence alignment, trimming and concatenation for the six loci of 169 taxa were done as described by Bank et al.[26]. We partitioned the supermatrix (4694 bp) in 12 subsets based on the three ribosomal genes (*16 S*, *18 S*, *28 S*) and the three codon positions of the three protein-coding genes (*COI*, *COII*, *H3*). The optimal partitioning scheme and best-fit substitution models under the corrected Akaike information criterion (-m MF–merge -merit AICc) were determined with IQ-TREE v.2.1.1[105–107], which kept all partitions separate except for the first codon position of the *COX* genes (see Supplementary Data 3). We used the resulting scheme to perform 50 independent tree searches under the Maximum Likelihood (ML) criteria and based on a random starting tree, smaller perturbation strength and an increased number of unsuccessful iterations before stopping (-p -t RAND -pers 0.2 -nstop 200). Node support was estimated from 10,000 ultrafast bootstrap trees (UFBoot[108]) and 300 standard non-parametric bootstrap (BS) trees, with the number of replicates being *a posteriori* determined as sufficient by the bootstopping criteria of RAxML v. 8.2.12[109]. Subsequently, we mapped the support values on the tree for which the highest log-likelihood had been calculated in IQ-TREE (-z option). We further assessed node support using the SH-aLRT single branch test[110] with 10,000 replicates and an independent tree search. Tree visualisation and rooting with Aschiphasmatidae were done in FigTree v.1.4.4 (https://github.com/rambaut/figtree).

**Divergence time estimation and historical biogeography**. Divergence times were estimated simultaneously with a Bayesian phylogenetic inference (BI) in BEAST v. 2.6.1[111]. While we used the same partitioning scheme as for the ML analysis, the optimal substitution model was selected by the bModelTest v.1.2.1[112] package implemented in BEAST. Trees and clocks were linked across all partitions employing the Yule tree prior and a relaxed molecular clock[113] with a clock rate of $1e^{-7}$ and assuming an uncorrelated lognormal distribution clock model (UCLD). In order to render Aschiphasmatidae as the outgroup, we constrained the remaining taxa as monophylum (=Neophasmatodea). Two runs were performed for 100 million MCMC generations sampled every 2,000 generations with different calibrations to explore the divergence time of Phylliidae. For the first, a fossil calibration was set applying a lognormal distribution (offset = 47; mean + stdev = 1.0) based on the leaf insect fossil *Eophyllium messelense*[11]. Since meaningful fossils are scarce among Euphasmatodea and the respective taxa are not included in our taxon sampling, we applied a secondary calibration derived from Simon et al.[33] for our second divergence time estimation and calibrated the root of the tree (Euphasmatodea) with a normal distribution (mean = 80.3; sigma = 6). After validating the convergence in Tracer v.1.7.1[114] and removing 10 and 15% burn-in from the *Eophyllium*- and Euphasmatodea-calibrated trees, respectively, maximum clade credibility trees were obtained for both from the tree posterior distributions in TreeAnnotator v. 2.6.0 (BEAST package[111]).

The ancestral range estimation was based on the fossil-calibrated BEAST tree and performed with BioGeoBEARS v.1.1.2[115,116] as implemented in R 3.5.3[117] following the instructions given by Bank et al.[26]. Although the Southeast (SE) Asian geographical range of the taxon therein coincides with the phylliid distribution, the distributional pattern differs in regard to the pacific islands east of Wallace's Line as well as to Australia. Hence, in addition to the five proposed areas therein, we subdivided the area defined as "Eastern SE Asia" by Bank et al.[26] into three, resulting in the following eight geographical areas: (A) Mainland Asia, (B) Malay Peninsula + Western Indonesia including Sumatra and Java, (C) Borneo + Palawan Island, (D) Wallacea, (E) Northern Philippine Islands, (F) Southern Philippine Islands, (G) Australasia and (H) Pacific. We allowed only single or two adjacent areas as well as the three areas that connected Sundaland and continental Asia (A + B + C), adding up to 22 ranges to be used in the analysis (see Supplementary Data 4).

**Species delimitation**. The available (voucher) material was morphologically inspected and identified and wherever possible, we included the type material to eliminate the possibility of misidentification. Regarding the non-type material, we compared the voucher specimen morphologically with the original type specimens. The inclusion of species of which sequence data are already available online allowed for the detection of potential misidentification by previous authors. Since molecular investigation has revealed the presence of several cryptic leaf insect species[38,40], we have conducted a species delimitation analysis using the tree-based approach bPTP[118] in addition to our morphological examination and the inclusion of type material. We used the ML tree as input on the web server (http://species.h-its.org/) and ran 100,000 generations with default settings.

**Reporting Summary**. Further information on research design is available in the Nature Research Reporting Summary linked to this article.

## Data availability

The authors declare that the data supporting the findings of this study are available within the supplementary information files. Newly generated sequence data were deposited in GenBank under the accession numbers MW686032–MW686200,

MW698871–MW698927, and MW703187–MW703369 (for more details, please refer to Supplementary Data 1). Supplementary Data 5 contains the final supermatrix including the partitioning scheme in nexus format that was used for all phylogenetic inferences.

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

## Acknowledgements

We are grateful to many individuals for supplying specimens for this study: Alexander Banko (Canada), Tim Bollens (Belgium), Joachim Bresseel (Belgium), Thomas Buckley (New Zealand), Oskar Conle (Germany), Jérôme Constant (Belgium), Zhiwei Dong (China), Detlef Größer (Germany), Thierry Heitzmann (Philippines), Frank Hennemann (Germany), Gideon Kakabin (New Britain, Papua New Guinea), Albert Kang (Malaysia), Bruno Kneubühler (Switzerland), Maxime Ortiz (France), Johnson Sau (Malaysia), Francis Seow-Choen (Singapore) and Yingtong Wang. We thank Thomas R. Buckley for the DNA sequences of Bacillinae samples 'BAE1' and 'STI10', and Katja Kramps for the DNA extract of the holotype of *Phyllium toboloense*. We also thank Doug Yanega (USA) for notes on best taxonomic practices and Bruno Kneubühler (Switzerland) for allowing us to use his images of live phylliids to show the beauty of these insects. This study was supported by the German Research Foundation (DFG grants BR 2930/3-1, BR 2930/4-1 and BR 2930/5-1 to Sven Bradler). We acknowledge the support by the Open Access Publication Funds of the University of Göttingen.

## Author contributions

S.Ba., S.Br. and R.T.C. designed the project. S.Br. supervised the overall research project. R.T.C. and S.L.T. collected the materials. R.T.C. conducted the morphological examination. S.Ba., K.H. and Y.L. generated the molecular data. S.Ba. performed the data analyses and prepared the figures. S.Ba. wrote the manuscript with contributions from S. Br. and R.T.C. All authors approved the final version.

## Funding

## Competing interests

The authors declare no competing interests.
