## [Peer Review File · Communications Biology]

Reviewers' comments:

Reviewer #1 (Remarks to the Author):

The authors provide the most comprehensive phylogenetic analysis of leaf insects (Phylliidae) conducted to date. The study represents a valuable advance in our understanding of the group, especially given the rich supplementary material. As such, I recommend the manuscript for publication provided that major revisions are made.

While the authors experimented with tree-building models in both BI and ML settings, it would be ideal to also include compositionally heterogeneous models, such as variants of the C or UDM models in IQ-Tree. While the authors did run a test in IQ-Tree to determine the best-fitting model, it seems that they only tested the standard site-homogeneous models, which often underperform when compared to site-heterogeneous models (Kapli et al. 2021). This problem is even more important given that the position of Phylliidae within Phasmatodea has been shown to vary depending whether compositionally homogeneous and heterogeneous models were used (cf. Simon et al. 2019; Tihelka et al. 2020).

The final alignment analysed by the authors and provided in the Supplementary Data consists of nucleotides. As they represent a merely four-letter alphabet, nucleotide alignments are more prone to saturation, which is a significant source of bias in molecular studies. To investigate the effect of mitigating heterogeneity on the recovered topology, alternative analyses using an amino acid alignment should be performed. A separate analysis excluding the third codon position of COI can also be conducted, as they suffer from saturation and can potentially bias phylogenetic analyses (e.g., Lin & Danforth, 2004; Inagaki & Roger 2006).

Fossil phasmatodeans are scarce, but the authors could calibrate more outgroup taxa, instead of using the ingroup fossil *Eophyllium* as the sole calibration point. If the authors deem this as not possible, this should be justified in the Methods. In the second molecular clock analysis, the authors used the dates from Simon et al. It is not obvious why this study was used as a benchmark when different analyses have arrived at considerably incongruent timescales of stick and leaf insect evolution.

The discussion of molecular clock results should only focus on the recovered age ranges and preferably discard the means. Posterior means and medians are not very informative and often wrong, while posterior intervals provide much more accurate estimates of divergence times and quantify the associated uncertainty (Warnock et al. 2017). Thus, the age ranges should be printed on the supplementary figures, instead of just the means.

I don't see what is "problematic" about Forni et al. 45 (line 244). Their estimated age range encompasses the Early Cretaceous, when large eudicots were already around. Granted, the earliest part of their estimates seems a bit too old, but when interpreting molecular clock estimates, the total confidence interval should be taken into account rather than mean ages (as discussed above).

The statement "...rainforests dominated by angiosperm trees probably arose during the Cretaceous–Cenozoic transition 61,62" has to be revised. In fact, floras dominated by angiosperms are known from the Cenomanian (earliest Late Cretaceous) onwards (e.g., Coiffard et al. 2012). Flowering plants became near-ubiquitous and dominated the tropics by the end of the Cretaceous (Crane & Lidgard, 1989; Lupia et al., 1999), not during the Cretaceous–Cenozoic transition.

The statement "leaf masquerade cannot have evolved at a time predating the angiosperm predominance" should be revised. In fact, there are examples of insects looking like leaves and other plant parts dating to the early Mesozoic and even Palaeozoic. Although a few of these are contentious, many others are very convincing (Wedmann 2010; Wang et al. 2012, 2018; Garrouste et al. 2016; Yang et al. 2012 and references cited therein). It should not be surprising that some insects mimicked plants before angiosperms, as there are some today that mimic gymnosperms.

I look forward to seeing the revised text.

References

- Coiffard, C., Gomez, B., Daviero-Gomez, V., & Dilcher, D. L. (2012). Rise to dominance of angiosperm pioneers in European Cretaceous environments. *Proceedings of the National Academy of Sciences*, 109(51), 20955-20959.
- Crane PR, Lidgard S. 1989. Angiosperm diversification and paleolatitudinal gradients in Cretaceous floristic diversity. *Science* 246: 675–678.
- Garrouste, R., Hugel, S., Jacquelin, L., Rostan, P., Steyer, J. S., Desutter-Grandcolas, L., & Nel, A. (2016). Insect mimicry of plants dates back to the Permian. *Nature communications*, 7(1), 1-6.
- Kapli, P., Flouri, T., & Telford, M. J. (2021). Systematic errors in phylogenetic trees. *Current Biology*, 31(2), R59-R64.
- Lin, C.P. & Danforth, B.N. (2004) How do insect nuclear and mitochondrial gene substitution patterns differ? Insights from Bayesian analyses of combined datasets. *Molecular Phylogenetics Evolution*, 30, 686–702
- Lupia R, Lidgard S, Crane PR. 1999. Comparing palynological abundance and diversity: implications for biotic replacement during the Cretaceous angiosperm radiation. *Paleobiology* 25: 305–340.
- Simon, S., Letsch, H., Bank, S., Buckley, T. R., Donath, A., Liu, S., ... & Bradler, S. (2019). Old World and New World Phasmatodea: phylogenomics resolve the evolutionary history of stick and leaf insects. *Frontiers in Ecology and Evolution*, 7, 345.
- Tihelka, E., Cai, C., Giacomelli, M., Pisani, D., & Donoghue, P. C. (2020). Integrated phylogenomic and fossil evidence of stick and leaf insects (Phasmatodea) reveal a Permian–Triassic co-origination with insectivores. *Royal Society open science*, 7(11), 201689.
- Wang, Y., Labandeira, C. C., Shih, C., Ding, Q., Wang, C., Zhao, Y., & Ren, D. (2012). Jurassic mimicry between a hangingfly and a ginkgo from China. *Proceedings of the National Academy of Sciences*, 109(50), 20514-20519.
- Wang, Y., Liu, Z., Wang, X., Shih, C., Zhao, Y., Engel, M. S., & Ren, D. (2010). Ancient pinnate leaf mimesis among lacewings. *Proceedings of the National Academy of Sciences*, 107(37), 16212-16215.
- Warnock, R. C., Yang, Z., & Donoghue, P. C. (2017). Testing the molecular clock using mechanistic models of fossil preservation and molecular evolution. *Proceedings of the Royal Society B: Biological Sciences*, 284(1857), 20170227.
- Wedmann, S. A brief review of the fossil history of plant masquerade by insects. *Palaeontographica (B)* 283, 175–182 (2010).
- Yang, H., Shi, C., Engel, M. S., Zhao, Z., Ren, D., & Gao, T. (2021). Early specializations for mimicry and defense in a Jurassic stick insect. *National Science Review*, 8(1), nwaa056.

Reviewer #2 (Remarks to the Author):

The manuscript, "A tree of leaves: Phylogeny and biogeography of the leaf insects (Phasmatodea: Phylliidae)," provides the first look into the phylogeny of this charismatic insect family. Overall, this work provides some much-needed insight into the relationships of these leaf-like insects. The past lumping of taxa into a single genus seems to have led to a paraphyletic genus. This work aimed to resolve these issues and revise the taxonomy accordingly. In addition to the phylogenetic analysis, this work presented a hypothesis to explain the current distribution of this unique family.

The academic merit of this work is defensible and I have very little in the way of recommendations. The following presents the minor questions I had while reviewing the manuscript.

1. The character sampling included six genes, but I was unable to see how complete the character

sampling was without looking at the .txt alignment file. Is there a way/room to put a table in that shows the taxa and which genes were sampled for each?

2. Could the missing data mentioned above influence the analysis looking at species delimitation?

3. The pie chart on Figure 3 has a 4% piece that represents sampled taxa that were not included in this study. Why were those specimens sampled but not included?

Dear reviewers,

We thank you for the time and effort you put into reviewing our manuscript. Please find our replies in **green** below.

Reviewer #1 (Remarks to the Author):

The authors provide the most comprehensive phylogenetic analysis of leaf insects (Phylliidae) conducted to date. The study represents a valuable advance in our understanding of the group, especially given the rich supplementary material. As such, I recommend the manuscript for publication provided that major revisions are made.

Thank you very much for the overall positive estimation of our study and for the helpful critical comments that we haven taken into full consideration in order to improve our contribution. Please see below for our responses in detail.

1. While the authors experimented with tree-building models in both BI and ML settings, it would be ideal to also include compositionally heterogeneous models, such as variants of the C or UDM models in IQ-Tree. While the authors did run a test in IQ-Tree to determine the best-fitting model, it seems that they only tested the standard site-homogeneous models, which often underperform when compared to site-heterogeneous models (Kapli et al. 2021). This problem is even more important given that the position of Phylliidae within Phasmatodea has been shown to vary depending whether compositionally homogeneous and heterogeneous models were used (cf. Simon et al. 2019; Tihelka et al. 2020).

The final alignment analysed by the authors and provided in the Supplementary Data consists of nucleotides. As they represent a merely four-letter alphabet, nucleotide alignments are more prone to saturation, which is a significant source of bias in molecular studies. To investigate the effect of mitigating heterogeneity on the recovered topology, alternative analyses using an amino acid alignment should be performed. A separate analysis excluding the third codon position of COI can also be conducted, as they suffer from saturation and can potentially bias phylogenetic analyses (e.g., Lin & Danforth, 2004; Inagaki & Roger 2006).

Thank you for these suggestions, however the article by Kapli et al. (2021) explicitly refers to “large scale phylogenomic studies” and how “large data sets” are in conflict regarding the resolved topologies. In contrast, our dataset only covers six genes per species of which three genes are ribosomal and cannot be translated into amino acids. In particular, H3 is way too conservative to provide any information on the amino acid level (its sequence being nearly identical between our investigated taxa), while on the nucleotide level, H3 provides a considerable phylogenetic signal. We nevertheless gave this approach a try and produced a concatenated alignment of amino acids for COI, COII and H3, resulting in less than 630 amino acid positions.

We performed two separate tree inferences in IQ-TREE based on the alignment partitioned by genes and searched for the best-fit model among homogeneous and heterogeneous (CXX models) (please see figures below). In both trees, multiple lineages of stick insects, which were repeatedly corroborated in past studies and thus appear undisputed (see in-figure colour code below), are recovered as polyphyletic. Consequently, we cannot at all trust these results in regard to the internal relationships of leaf insects (and beyond) and must conclude that the amount of data and the choice of genes in our study is not convenient for phylogenetic analyses on the amino acid level. Since the site-heterogeneous models in IQ-TREE are only available for amino acid datasets (to the best of our knowledge), we are not able to fit models to test for compositional heterogeneity. In summary, given the unsatisfactory performance of this additional approach, we would prefer to omit this additional analysis from our contribution.

2. This problem is even more important given that the position of Phylliidae within Phasmatodea has been shown to vary depending whether compositionally homogeneous and heterogeneous models were used (cf. Simon et al. 2019; Tihelka et al. 2020).

While it is true that the phylogenetic position of Phylliidae among Phasmatodea is still under debate, our study was not designed at all to tackle this question, but to infer the internal relationship among leaf insects.

3. Fossil phasmatodeans are scarce, but the authors could calibrate more outgroup taxa, instead of using the ingroup fossil *Eophyllum* as the sole calibration point. If the authors deem this as not possible, this should be justified in the Methods. In the second molecular clock analysis, the authors used the dates from Simon et al. It is not obvious why this study was used as a benchmark when different analyses have arrived at considerably incongruent timescales of stick and leaf insect evolution.

Our outgroup sampling is quite comprehensive given the focus of the paper: deciphering the internal leaf insect phylogeny. However, by restricting our sampling to Euphasmatodea, neither Timematodea (its sister taxon) nor Embioptera (the sister group of Phasmatodea) are included, which would allow for further meaningful old fossils to be used for calibration. For euphasmatodean lineages, besides *Eophyllum*, there are only very few young fossils (eggs from Dominican amber, around 20 mya) available that were previously used (Robertson et al.²⁵, Simon et al.³³) and led to similar divergence times as recovered here. Since the respective taxa (i.e., *Malacomorpha*, *Clonistria*) are not included in our taxon sampling, we chose to use a secondary calibration based on the divergence times estimated by Simon et al. (2019). We have modified this section in order to explain this in the M&M (line 405):

“Since meaningful fossils are scarce among Euphasmatodea and the respective taxa are not included in our taxon sampling, we applied a secondary calibration derived

from Simon et al.³³ for our second divergence time estimation and calibrated the root of the tree (Euphasmatodea) with a normal distribution (mean = 80.3; sigma = 6).“

The discussion on the choice of fossils and the resulting discrepancies among studies is admittedly short, owing to word restrictions for articles by Communications Biology (“The choice of unequivocal fossils and appropriate calibration points is essential and their inconsistent application may lead to substantial discrepancies among studies on phasmatodean evolution (but see previous discussions^{10,26,49})“), but refers to papers that discuss the use of unambiguous fossils in more detail. For instance, Bank et al.²⁶ states:

“Tihelka et al. (2020) included fossils that were intentionally excluded in the study by Robertson et al. (2018), such as *Echinosomiscus primoticus* Engel & Wang, a fossil insect preserved in Cretaceous amber (~99 mya) described as an adult male related to a subordinate lineage comprising Lonchodinae and Clitumninae (Engel et al., 2016). However, this extremely small fossil most probably does not belong to Phasmatodea at all (Bradler & Buckley, 2018) and was used as calibration point for Phasmatodea or Euphasmatodea (Simon et al., 2019; Forni et al., 2020; Forni et al., 2021), whereas Tihelka et al. (2020) included it as calibration point within the much more subordinate Oriophasmata. Another important fossil specimen included by Tihelka et al. (2020) is a Jurassic heelwalker (Mantophasmatodea) described by Huang et al. (2008) that needs to be critically reassessed. Bradler & Buckley (2011) emphasized the importance of rigorously interpreted and unambiguously placeable fossils as reliable calibration points on phylogenetic trees.”

4. The discussion of molecular clock results should only focus on the recovered age ranges and preferably discard the means. Posterior means and medians are not very informative and often wrong, while posterior intervals provide much more accurate estimates of divergence times and quantify the associated uncertainty (Warnock et al. 2017). Thus, the age ranges should be printed on the supplementary figures, instead of just the means.

We depict all mean values together with their ranges in the text and therefore do not see a problem with this way of presentation (we have modified the two incidences, where we forgot to include the range). We have added the approximate range for the age estimate by Forni et al.⁴⁵ in line 251 (also see below).

As for the supplementary figures: We do not depict the mean ages and only show the confidence intervals in these figures. The exact numbers (means and ranges) can be found in the nexus tree file (Supplementary Data).

5. I don't see what is “problematic” about Forni et al.⁴⁵ (line 244). Their estimated age range encompasses the Early Cretaceous, when large eudicots were already around. Granted, the earliest part of their estimates seems a bit too old, but when

interpreting molecular clock estimates, the total confidence interval should be taken into account rather than mean ages (as discussed above).

Thank you for the suggestion. In fact, our statement was a bit harsh and unspecific here. We have modified the sentence now, more strongly emphasizing the range (line 250):

“In particular, large parts of the lower ages estimated by Forni et al.⁴⁵ (approximately 170–90 mya) appear to be too old given that eudicot angiosperms are hypothesised to have been subordinate herbs until the mid-Cretaceous^{55,61}, a span of time only covered by the upper confidence interval in Forni et al.’s study⁴⁵.”

6. The statement “...rainforests dominated by angiosperm trees probably arose during the Cretaceous–Cenozoic transition^{61,62}” has to be revised. In fact, floras dominated by angiosperms are known from the Cenomanian (earliest Late Cretaceous) onwards (e.g., Coiffard et al. 2012). Flowering plants became near-ubiquitous and dominated the tropics by the end of the Cretaceous (Crane & Lidgard, 1989; Lupia et al., 1999), not during the Cretaceous–Cenozoic transition.

Thank you for pointing this out, the word “transition” was indeed not reflecting what we intended to say, since we meant to describe the time period around this transition (as depicted in Figure 4). We revised the sentence as follows and added the two references suggested by the reviewer (line 254): “The first forest trees may have occurred from that time on, but rainforests dominated by angiosperm trees probably arose at the end of the Cretaceous^{61–64}.”

7. The statement “leaf masquerade cannot have evolved at a time predating the angiosperm predominance” should be revised. In fact, there are examples of insects looking like leaves and other plant parts dating to the early Mesozoic and even Palaeozoic. Although a few of these are contentious, many other are very convincing (Wedmann 2010; Wang et al. 2012, 2018; Garrouste et al. 2016; Yang et al. 2012 and references cited therein). It should not be surprising that some insect mimicked plants before angiosperms, as there are some today that mimic gymnosperms. I look forward to seeing the revised text.

Thank you for your suggestions! We have added information regarding pre-angiosperm leaf mimicry in stick insects and specified that Phylliidae are uniformly imitating angiosperm leaves (line 236):

“While it had been argued before that leaf mimicry predated the more common twig mimicry of extant forms, since fossil stem-Phasmatodea as well as members of Timematodea, the sister taxon of Euphasmatodea, exhibit leaf mimicry^{14,52}, it appears undisputed that phylliid leaf insects derived from twig-imitating forms¹¹ and secondarily evolved angiosperm leaf imitation more recently.”

To further clarify that we are referring to the imitation of angiosperm leaves and not leaves in general, we also modified the following sentence (line 256): “Interestingly,

the origin of other angiosperm leaf-mimicking insects such as members of the orthopteran Tettigoniidae^{17,65} or the *Kallima* butterflies^{16,66} appear to coincide with our age estimates for Phylliidae, supporting our claim that leaf masquerade involving angiosperm leaf imitation cannot have evolved at a time predating the angiosperm predominance.”

Reviewer #2 (Remarks to the Author):

The manuscript, “A tree of leaves: Phylogeny and biogeography of the leaf insects (Phasmatodea: Phylliidae),” provides the first look into the phylogeny of this charismatic insect family. Overall, this work provides some much-needed insight into the relationships of these leaf-like insects. The past lumping of taxa into a single genus seems to have led to a paraphyletic genus. This work aimed to resolve these issues and revise the taxonomy accordingly. In addition to the phylogenetic analysis, this work presented a hypothesis to explain the current distribution of this unique family.

The academic merit of this work is defensible and I have very little in the way of recommendations. The following presents the minor questions I had while reviewing the manuscript.

Thank you very much for your positive estimation. We have addressed your suggestions in detail below.

1. The character sampling included six genes, but I was unable to see how complete the character sampling was without looking at the .txt alignment file. Is there a way/room to put a table in that shows the taxa and which genes were sampled for each?

This information is presented in detail in Supplementary Table 2, which is too extensive to be incorporated into the main text, but we have added a summarising sentence in the Results & Discussion section now (line 107) (and accordingly changed the numbering of the Supplementary Tables 1 and 2):

“For 77% of all analysed taxa, we obtained the sequences of five or six genes and only for 3% of the included taxa we could generate sequences of one or two genes (for further details see Supplementary Table 1).“

2. Could the missing data mentioned above influence the analysis looking at species delimitation?

The species delimitation analysis is based on solely the tree and thus does only consider the branch lengths and number of substitutions. Therefore, missing data cannot have a direct influence on the analysis.

3. The pie chart on Figure 3 has a 4% piece that represents sampled taxa that were not included in this study. Why were those specimens sampled but not included?

The data for these four species (that all pertain to the herein well sampled genus *Cryptophyllum*) were largely contributed by project partners not co-authoring and not involved in the present study and after the analyses for the current study were accomplished. These unsampled taxa have just been published in a taxonomic study by Cumming et al. (2021)⁴⁰, and since those surely have no influence on the outcome of the phylogeny presented here, we decided not to rerun all analyses for these minor and negligible additions. Nevertheless, we did not want to ignore the existence of these taxa in our overview in Fig. 3.

Figure B. Phylogenetic relationships based on concatenated amino acid alignments of COI, COII and H3 with the latter having no parsimony-informative sites. Partitioned by gene with best-fit model among the tested site-heterogeneous models (C10-C60) + gamma rate and FreeRate heterogeneity in IQ-TREE (COI: C60+G4+R5; COII: C60+G4+R5; H3: C20+G4+R2).

REVIEWERS' COMMENTS:

Ref 2:

Thank you for making these clarifications and edits. I wholeheartedly recommend the manuscript for publication in its present state.